# The Dialogical Turn in Normative Political Theory and the Pedagogy of Human Rights Education

**Dale T. Snauwaert**

Department of Educational Foundations and Leadership, Judith Herb College of Education, The University of Toledo, 2801 W Bancroft St, Toledo, OH 43606, USA; dale.snauwaert@utoledo.edu

**Abstract:** The purpose of this paper is to explore a capacity-building pedagogical approach to human rights education as a complement to the "declarationist" approach. The basic premise of this philosophical paper is the idea of human rights as justified claims and/or demands; as such, ethical and moral justification is presupposed in the very idea of rights itself. It is argued that a *dialogical turn* in moral and political philosophy, in particular theoretical justifications of principles of justice, such as rights, has taken place. Given that ethical and moral justification is central to the meaning of human rights, the significance of this dialogical turn for the idea of human rights *and* human rights education is explored from within the idea of the logical structure of disciplines of knowledge, a discipline's fundamental ideas and forms of thought (methods of inquiry). From within this perspective, it is argued that the dialogical nature of justification central to rights should structure the pedagogy of human rights education. It is suggested that this pedagogy entails three forms of normative dialogue—ethical, moral, and critical—that can form the normative structure of a pedagogy of human rights education. It is concluded that while awareness and respect are necessary conditions to the realization of human rights, the development of the capacity of future citizens to make, to justify, and to critique human rights claims is also necessary for the realization of human rights.

**Keywords:** human rights; human rights education; ethical and moral justification; political philosophy; dialogue; dialogical turn

## 1. Introduction

A "declarationist" approach to human rights education has been dominant in the history of the idea and practice of human rights education. This approach has focused on the cultivation of an awareness of and respect for human rights. While awareness and respect are necessary conditions to the realization of human rights, it can be argued that the development of the capacity of future citizens to make, to justify, and to critique human rights claims is also necessary [1]. This perspective adds a capacity-building element to the idea of human rights education, with particular pedagogical implications. As a means to the further development of this capacity-building element, a conception of human rights as justified claims and/or demands is articulated as a basic premise, and as such, it is argued that ethical and moral justification is presupposed in the very idea of rights. Second, it is argued that a *dialogical turn* in moral and political philosophy, in particular theoretical justifications of principles of justice, such as rights, has taken place. Third, given that ethical and moral justification is central to the meaning of human rights, the significance of this dialogical turn for the idea of human rights *and* human rights education is explored from within the idea of the logical structure of disciplines of knowledge, a discipline's fundamental ideas and forms of thought (methods of inquiry). From within this perspective, the dialogical nature of justification central to rights should structure the pedagogy of human rights education. It is suggested that this pedagogy entails three forms of normative dialogue—ethical, moral, and critical—that can form the normative structure of a pedagogy

of human rights toward the outcome of human rights capacity development—the capabilities necessary for citizens to realize their own and others' human rights. This paper is a work of philosophy and employs the standard methods of philosophical inquiry. As such, the paper does not make empirical claims nor are empirical findings offered. The purpose of the paper is to outline a philosophical conception of a human rights pedagogy.

## 2. Basic Premise: Human Rights as Justified Claims

Human rights can be conceived as what a human being is justified in demanding and thus claiming as a matter of right/justice. As Henry Shue suggests:

> A moral right provides (1) the rational basis for a justified demand and (2) that the actual enjoyment of a substance be (3) socially guaranteed against standard threats. [2] (p. 13)

> Basic rights, then, are everyone's minimum reasonable demands upon the rest of humanity. They are the rational basis for justified demands the denial of which no self-respecting person can reasonably be expected to accept. [2] (p. 19)

From this perspective, rights are justified demands for the enjoyment of social goods, which are guaranteed by the society. A right is a *rational* basis for a *justified* demand in the sense that it provides compelling *reason(s)* for the demand being met. Moreover, it is a demand based in moral equality/human dignity [2]. As demands, rights have to do with the activity of claiming—to demand something is to assert claim onto others; claiming in turn is a rule-governed activity: "To have a claim . . . is to have a case meriting consideration . . . to have reason or grounds that put one in a position to engage in performative and propositional claiming" [3] (p. 185). As such, human rights are "moral claims upon the organization of society" [4] (p. 200); in turn, the organization of society is founded upon a conception of justice that regulates its basic structure [5,6].

The validity of a rights claim is contingent upon justification within a system of rules. Therefore, legal rights are valid claims within a system of law, whereas a moral rights claim is valid in terms of moral principle. From this perspective, respect for persons is constituted by recognizing another as a potential maker of claims. Rights are, therefore, *justified moral* and, when codified into law, *legal claims*.

Furthermore, the substantive content of human rights (what one has a human right to) is comprised of ethical goods. An ethical good is that which we have reason to value, such as freedom, political self-determination, security of person, due process, subsistence, education, health care, etc. To assert that one should have a right to some good is to claim that that good possesses such value that it stands above mere preference, that it is fundamentally important to the pursuit of a good life. This claim of the value of the good also requires normative justification, which entails both ethical and moral justification (discussed below).

Justification is therefore presupposed within the very idea of a right. Justification pertains to both the content of the right as well as its general justifiability as a claim. We can ask: what is the rational basis of the justification of claims as matters of right? This question defines the pursuit, in part, of moral and political philosophy. As argued below, philosophical normative justification has taken a dialogical turn.

## 3. The Dialogical Turn in Moral and Political Philosophy

In the second half of the 20th century and the beginning of the 21st century, a *dialogical turn* in moral and political philosophy, including approaches to the normative justification, has emerged [7]. Given that moral justification is central to the meaning of human rights, this dialogical turn is very significant for the idea of human rights *and* the pedagogy of human rights education (This section is a revised version of a part of the review essay [6] with permission from *In Factis Pax.*).

The two most prominent modern (enlightenment) moral theories, Utilitarianism and Kant's deontological theory, proceed from a subjectivist orientation. Utilitarianism defines moral rightness in

terms of the maximization of aggregative utility. It posits that what is right is the action and/or rule that maximizes the best overall consequences. It measures overall consequences by the ranking of individual utility (by some measure of utility, such as pleasure or preference satisfaction) aggregated into a total sum of utility. The highest sum-ranked, average aggregate utility caused by the action or rule is that which is just. The utilitarian calculation is thus based in the equal consideration of individual subjective states of affairs. Although quite different in many ways, Kant's deontological theory is also subjectivist. He maintains that in the process of moral justification "... we merely make reason attend ... to its own principles." [8] (p. 404). In other words, the criteria of the justifiability and validity of moral norms can be constructed from within the presuppositions of reasonable individual moral judgment, that is, solely within the reason of the individual alone—a process of internal subjective reflection.

Subsequently, a prominent shift in moral and political philosophy has emerged, which constitutes a shift from a subjective to an intersubjective orientation; this shift entails a significant dialogical turn in the sense that dialogue has come to be understood as the central process of normative justification. Justification is an inherent feature of reason, for by definition justification entails the offering of reasons in support of one's claim. It is recognized that the hallmark of human reason—theoretical, practical, and instrumental—is that the validity of any claim is grounded in intersubjective mutual agreement [9–12]. Reason is not solely subjectively internally focused—it is directed outward toward others. This is true of normative justification. As the moral philosopher Rainer Forst maintains: "Respect for others does not rest on my relation to myself as 'making laws for myself' but corresponds to an original duty toward others that must be 'apprehended' and 'acknowledged' at the same time" [13] (p. 55). The existence of others in an intersubjective relationship demands "a 'response-ability' toward, not for, others . . . 'the other' represents for me an "unconditional" call to responsibility, one that I can reject only at the cost of violating morality. . . . it is the "face" of the other that makes clear to me where the ground of being moral lies" [13] (p. 59). It is the other person that obligates one to treat that person with respect, and respect structures a duty to offer them justification for the basic moral norms that will govern their relationship as persons.

This intersubjective call of the other is the basis of the dialogical turn in various approaches to moral and political philosophy: deontological moral constructivism, communitarianism, Walzer's interpretative approach, and capabilities theory, among others. A brief summary of each will highlight the dialogical turn within them.

### 3.1. Deontological Moral Constructivism

Moral constructivism refers to a process in which moral norms are justified through a procedure of deliberation that is structured and defined in terms of fairness, such that the fair conditions of agreement between the parties "represent what citizens would adopt in a situation that is fair between them" [14] (p. 310). The moral legitimacy of the results of deliberation (the mutual agreement) are transferred from the normative validity of the elements of fairness to what is agreed to. "The fairness of the circumstances under which agreement is reached transfers to the principles of justice agreed to ... What is just is defined by the outcome of the procedure itself" [14] (pp. 310–311). The validity of the moral principles of justice agreed to is in a sense "constructed" through a fair procedure of dialogical intersubjective justification [5,10,13]. In this way, Kant's subjective constructivist procedure is reconstructed in intersubjective dialogical terms. The validity of the justification of moral claims, such as human rights, rests upon intersubjective mutual agreement achieved through dialogue under fair conditions. From this perspective, valid moral norms and ethical values rest upon sharable reasons exchanged in a deliberative, dialogical process [10,13,15–18]. As Thomas Scanlon suggests: "thinking about right and wrong is, at the most basic level, thinking about what could be justified to others on grounds that they, if appropriately motivated, could not reasonably reject" [19] (p. 197).

### 3.2. Communitarianism

Communitarians maintain that normative justification can *only* be grounded in a substantive collective ethical identity, in the sense that what is morally right is derived from and justified by shared ethical values. Communitarians maintain that individual identity is ontologically grounded in culture. They assert a dialogical understanding of identity as formed in the context of comprehensive conceptions of the good life implicit in the culturally thick traditions of various communities [20,21]. They maintain that the moral right dialogically emerges out of, and is thus grounded in, the web of human relationships which constitute communal life. For example, Michael Sandel articulates the idea of a "constitutive community", constitutive in the sense that the collective ethical identity of the community forms the ethical identity of its members. Identity is not chosen, it is found within the ethical values of the community. In turn, the members' obligations are situated within the ethical identity of the community [20,22]. Alasdair Macintyre also maintains that the identity of persons is dialogically formed within the context of the community, in particular in the social "roles" that define communal life and one's place in it that in turn contain obligations which one can only reject at the loss of identity. From this perspective, a coherent conception of morality is based in a conception of the human good [23]. Valid justification of moral norms is based upon collectively shared ethical values forged out of communal dialogical relationships.

### Michael Walzer's Interpretative Approach

Walzer maintains that ethical and political philosophy proceeds intellectually by the application of an interpretative method [24–27]. Consistent with the communitarian perspective, morality is neither discovered in the fabric of reality (e.g., religious ethics, natural law ethics), nor is morality invented, that is, constructed through procedural processes (moral constructivism) [5,28]. Walzer argues that we do not have to discover or invent the moral world, it already exists; we are situated within it. Our own cultural traditions are the ultimate sources of morality; we do not need to discover or invent it, we need to interpret it. There is no other starting point for moral speculation; we start from where we are, reflecting on actually existing ethical beliefs. This reflection entails *dialogue* about the *meaning* of ethical goods and values. Fidelity to the deepest meaning of our most cherished values uncovered through a dialogical process of interpretation is the ethical standard of justification.

### 3.3. Capabilities Theory

In Amartya Sen's capabilities theory of justice, what is just is defined as that which promotes the realization of the combined index of capabilities of members of society as determined by the methods of social choice theory, comparative assessment, open impartial scrutiny, and public reasoning [19]. In other words, the state of affairs that ranks highest in terms of the combined index of capabilities is the most just among comparative alternatives. The process of comparative assessment proceeds for Sen through public reasoning—open and informed public deliberation, which tests the validity of the assessment in terms of the capacity to survive informed open impartial scrutiny through the exercise of public deliberation and public reason. Given the centrality of public reason for the comparative assessment of justice, there is an intimate connection between justice and democracy. Democracy is understood by Sen as public reasoning and deliberation. The pursuit of justice therefore can only proceed in terms of open impartial dialogue among citizens in the exercise of their public reason.

These examples highlight a significant *dialogical turn* in various methodologies of normative justification, placing dialogue at the center of ethical and moral justification. As argued below, this dialogical turn is highly significant for the theory and practice of human rights education.

## 4. The Logical Structure of Disciplines and the Pedagogy of Human Rights Education

As noted above, the purpose of this paper is to explore a capacity-building approach to human rights education as a complement to the "declarationist" approach. While awareness and respect are

necessary conditions for the realization of human rights, the development of the capacity to make, to critique, and to justify human rights claims is also fundamental for the realization of human rights. This perspective adds a capacity-building element to the idea of human rights education. To explore these pedagogical implications, the idea of the logical structure of disciplines of knowledge and inquiry is used here as a framework. This framework is based upon a curricular and instructional approach that places the teaching emphasis on the discipline's forms of thought. The forms of thought are instantiated in the *methods of inquiry characteristic of the discipline* [29–33]. The teaching focus is on the way members of the discipline systematically pursue knowledge, including standards of justification.

From this perspective, if human rights are understood as justified claims, thereby placing ethical and moral justification at their core, and if justification is dialogical in its basic structure, then dialogical justification should be at the core of the teaching approach to human rights education. A pedagogy consistent with the dialogical structure of ethical and moral justification of human rights, therefore, can be formulated as a dialogical pedagogy of the practice of justification.

There are at least three forms of dialogical justification that pertain to human rights: ethical dialogue, moral dialogue, and critical dialogue [10]. These forms of normative dialogical discourse can be employed with students as pedagogies within human rights education.

The ethical justification of particular goods as the content of human rights is contingent upon their role in defining the collective self-understanding and identity of the social group. Ethical goods constitute our self-understanding of who we are and want to be. The goods that are most central to this self-understanding constitute the ethically justifiable content of human rights claims. This process entails the hermeneutic interpretation of the meaning and value of the good grounded in the collective identity of the social group one is asserting the claim to [34]. This interpretation requires public deliberation and dialogue in order to reach a shared understanding. This dialogical hermeneutic process is an important pedagogical element of human rights education [1,35].

However, it is important to note that within culturally diverse communities the viability of a common collective ethical identity upon which valid ethical justification can be based may be problematic, in the sense that there is a potential danger of the imposition of the ethical values of one group onto others. This point suggests that, in addition to the ethical justifiability of the content of rights, the *moral justification* of the ethical goods/values that constitute that content is also required. Ethical values, therefore, must be generally accepted as morally valid, and that validity is contingent upon general moral justification [34].

As discussed above, the validity of moral justification rests upon intersubjective mutual understanding and agreement concerning moral principles *under fair conditions*. A human right as a justified demand is valid if it can be mutually agreed to under fair conditions. Valid moral norms must pass the test of fairness, resting on valid, sharable reasons. The process of justifying moral norms goes through a procedure of deliberation that is structured and defined in terms of *fairness*, such that the fair conditions of agreement between the parties "represent what citizens would adopt in a situation that is fair between them" [14] (p. 310). The moral justifiability of the values and principles agreed to as a result of deliberation between those affected is derived from the normative validity of the criteria of fairness that regulates the deliberation. Thus, the moral justifiability of human rights claims entails a dialogical process of deliberation that seeks fair mutual agreement about the validity of making claims as well as the content of those claims. Students of human rights can thus be engaged in this process of moral justificatory discourse at the core of the pedagogy of human rights education.

Furthermore, critical dialogue and a critical dialogical pedagogy should also be considered central to human rights education. Those who seek to deny the human rights of others through the pursuit of domination and oppression often employ invalid, ideological justifications that attempt to justify human rights denials and abuses. Unjust social structures and practices are sustained by invalid justifications and patterns of thought that constitute "cultural violence . . . the symbolic sphere of our existence ... that can be used to justify or legitimize direct or structural violence . . . and thus [rendering it] acceptable in society" [36] (pp. 291–292). The validity of these justifications fails to survive critical

scrutiny. One approach to the propagation of invalid justifications is the delegitimization of truth, often expressed as hostility to the process of verification; in extreme cases, even the very idea of truth itself is called into question. Furthermore, accompanying efforts to delegitimize institutions that promote and sustain independent thought, in particular universities and the free press, are continually pursued, which in turn undermines the value of expertise as a source of truth. This attack on truth and its institutions leads to the delegitimization of valid moral and legal norms, including rights. This normative delegitimization degrades and debases public deliberation, turning it into mere sloganeering and appeals to fear and anger, rather than reasoned argument [34,37–40].

*Critical dialogue* involves the critique of invalid justifications which serve as ideological, culturally violent justifications for human rights abuses and denials. Critical dialogue further involves an understanding and critique of the functioning and impact of power within political institutions and knowledge and critical analysis of the structures of society. Therefore, from this perspective, the pedagogy of human rights education should also engage students in the processes of critical dialogue around invalid justifications of human rights denials and abuses [34].

## 5. Conclusions

Based upon the premise that human rights can be understood as justified claims, it was argued that ethical and moral justification is presupposed in the very idea of rights. It was then argued that a *dialogical turn* in moral and political philosophy, in particular theoretical justifications of principles of justice, such as rights, has taken place. Given that ethical and moral justification is central to the meaning of human rights, the significance of this dialogical turn for the idea of human rights *and* for a pedagogy human rights education was explored from within the idea of the logical structure of disciplines of knowledge. From within this perspective, it can be concluded that the dialogical nature of ethical and moral justification central to rights should structure the pedagogy of human rights education and that this pedagogy entails three forms of normative dialogue—ethical, moral, and critical—that together form the normative structure of human rights education. A pedagogy of ethical, moral, and critical justification should be considered as a means of developing the capacity of future citizens to make, to justify, and to critique human rights claims.

**Funding:** This research received no external funding.

**Conflicts of Interest:** The author declares no conflict of interest.

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
