# Peer review of "The Dialogical Turn in Normative Political Theory and the Pedagogy of Human Rights Education"

_education, doi:10.3390/educsci9010052_

Reviewer 1 Report

Fascinating and very timely discussion on invalid justifications and delegitimization of truth. I would have liked to see more analysis on how this affects or should be addressed in the pedagogy of human rights education. I think your conclusion is short in addressing the significance of your concerns and insights. Please correct minor misspellings and typos. 

I recommend revision to highlight the author's analysis of challenges faced by human rights education and possibly adjusting the title of the article.

Author Response

Thank your for your helpful comments.  I've some substantive revisions to the paper clarifying the focus of paper on the pedagogy of human rights education.  I believe both the introduction and conclusion are now better framed.  I also revised the title to better reflect the focus of the paper. I've fixed the typos.  I've attached the revised manuscript with track changes.

Reviewer 2 Report

This is a very interesting piece, which I think has potential to become an important contribution to both human rights philosophy and the literature on human rights education. 

I have only a few comments that the author might want to consider when revising this for publication. 

1) The part on human rights as justified claims would benefit from greater clarity. Some distinctions – e.g. claims vs. claim-making, and claims to vs. claims against someone (see Joel Feinberg) –  are not sufficiently clear. Also, I think it's worthwhile mentioning that the understanding of human rights as justified claims remains a contested position within the field of human rights philosophy. I think the piece would benefit from a more elaborate argument as to why this notion is better than, for instance, the political approach that scholars like Charles Beitz derive from John Rawls's writings on the laws of nations. It also might be worth considering more recent interventions in the field of human rights philosophy, including the recent edited volume by Rowan Cruft, S. Matthew Liao, and Massimo Renzo. I also lack a reference to James Griffin's seminal work.

2) As I see it, the principal contribution of this piece is that it connects the philosophical discourse on justification with the topic of human rights education. However, it is not really clear what the author thinks that human rights education is all about. The author doesn't seem to distinguish between "human rights education" and "human rights studies." While the former can take many forms, it ultimately aims to promote greater awareness of and respect for human rights norms. This is different from human rights studies, which is the critical, scholarly study of human rights concepts and practices. I think it might be worthwhile clarifying the concepts here, and also connect the piece to more recent scholarly engagements with "human rights education" (see., e.g. Alexandre Lefebvre, Human Rights and the Care of the Self)

3) The piece is fairly short. This might not be a problem for a theoretically oriented paper. However, I think all sections could benefit from some expansion. I think the connection to the current state of play (e.g. fields of human rights philosophy and human rights education) should be strengthened. Above all, the author's main contributions to these fields should be spelt out more clearly. To what extent is the author's perspective and conclusions different from what has already been written and published in these areas? 

All this notwithstanding, I think this is a great article and I recommend that it should be published. 

Author Response

Thank your for your helpful and insightful comments.  

Regarding your point about rights as claims, I appreciate your suggestion to expand the discussion, however, I do not have time now to make such substantial revisions--it would constitute a different paper.  I therefore framed the idea of rights as claims as a basic premise, an assumption, on the basis of which the argument is grounded.

I've made some substantive revisions to the paper clarifying the focus of paper on the pedagogy of human rights education.  I framed the discussion in terms of adding a complement to the declarations approach to human rights, a complement that highlights the need for an additional capacity-building element to human rights education, in particular its pedagogy.

I believe both the introduction and conclusion are now better framed.  

I also revised the title to better reflect the focus of the paper.  

I've attached the revised manuscript with track changes.
